# The Influence of Periodontal Diseases and the Stimulation of Saliva Secretion on the Course of the Acute Phase of Ischemic Stroke

**DOI:** 10.3390/jcm11154321

**Published:** 2022-07-25

**Authors:** Wioletta Pawlukowska, Bartłomiej Baumert, Agnieszka Meller, Anna Dziewulska, Alicja Zawiślak, Katarzyna Grocholewicz, Przemysław Nowacki, Marta Masztalewicz

**Affiliations:** 1Department of Neurology, Pomeranian Medical University, 71-252 Szczecin, Poland; agoschorska@gamil.com (A.M.); nowackiprz@gmail.com (P.N.); kkneurol@pum.edu.pl (M.M.); 2Department of Hematology and Transplantology, Pomeranian Medical University, 71-252 Szczecin, Poland; bbaumert@pum.edu.pl; 3Department of Interdisciplinary Dentistry, Pomeranian Medical University, 70-111 Szczecin, Poland; anna.dziewulska@yahoo.com (A.D.); alicja.zawislak@pum.edu.pl (A.Z.); zstomaog@pum.edu.pl (K.G.); 4Department of Maxillofacial Orthopedics and Orthodontics, Institute of Mother and Child, 01-211 Warsaw, Poland

**Keywords:** saliva, stroke, periodontitis, saliva stimulation

## Abstract

Background and purpose: The course of an ischemic stroke depends on many factors. The influence of periodontal diseases and the stimulation of salivation on the course and severity of stroke remains unresolved. Therefore, the aim of the study was to analyze the severity of ischemic stroke depending on the occurrence of periodontal diseases and saliva stimulation. Methods: The severity of the neurological condition was assessed using the NIHSS scale on days one, three and seven of stroke. The incidence of periodontal diseases was classified using the Hall’s scale in the first day of stroke. On days one and seven of stroke, the concentration of IL-1β, MMP-8, OPG and RANKL in the patients’ saliva was assessed using the Elisa technique. At the same time, the level of CRP and the number of leukocytes in the peripheral blood were tested on days one, three and seven of the stroke, and the incidence of upper respiratory and urinary tract infections was assessed. Results:100 consecutive patients with their first ever ischemic stroke were enrolled in the study. 56 randomly selected patients were subjected to the stimulation of salivation, the remaining patients were not stimulated. In the study of the severity of the neurological condition using the NIHS scale on days three and seven of stroke, the degree of deficit in patients without periodontal disease significantly improved compared to patients with periodontal disease, respectively (*p* < 0.01 and *p* = 0.01). Patients from the stimulated group had more severe neurological deficit at baseline (*p* = 0.04). On days three and seven of neurological follow-up, the condition of patients from both groups improved with a further distinct advantage of the unstimulated group over the stimulated group, respectively (*p* = 0.03 and *p* < 0.001). In patients from both groups, a statistically significant decrease in CRP and lymphocyte levels was observed on day seven in relation to day one. Conclusions: The occurrence of periodontal disease in a patient with stroke affects the severity of stroke. Stimulation of the mouth and salivary glands in these patients may have a positive effect on the course of stroke, taking into account the dynamics of neurological symptoms.

## 1. Background

Stroke is one of the most common causes of in-hospital death. It is estimated that 11–15% of patients die from ischemic stroke in hospitals [1,2,3,4]. A direct correlation has been demonstrated between the extent of the stroke and the intensity of pro-inflammatory mechanisms in the area of acute focal cerebral ischemia and disorders in the immune system [5,6,7]. Patients with more severe stroke, i.e., with greater neurological deficits, are additionally exposed to infections due to immobilization, catheterization of the urinary tract and prolonged venous cannulation [8,9,10].

It has been proven that poor oral health may be an additional factor that increases the susceptibility to infections in patients with stroke [11,12]. Periodontitis is a chronic, infectious disease of the oral cavity, which occurs in about 85% of the human population in the world. It includes gingivitis with subsequent bone involvement of the alveolar processes and tooth loss. Biofilm is considered as etiological factor in the onset of gingivitis and it is implicated in the progression to periodontitis and peri-implant inflammation. Host and environmental factors significantly contribute to the advancement of the inflammatory process [12]. Studies have shown that periodontitis is clustered with low serum vitamin D levels [13]. In most cases, it is initiated by plaque and tartar bacteria, accumulating as a result of insufficient oral hygiene [14].

Periodontal disease is characterized by chronic inflammation, induced by pro-inflammatory cytokines, chemokines, and other mediators. IL-1β is a key cytokine that plays a significant role in the induction and maintenance of an inflammatory response in periodontal tissue. Its concentration in saliva can be treated as a marker of the intensity of the inflammatory process in the oral cavity [15,16].

Another well-known biomarker of periodontitis is metalloproteinase-8 (MMP-8), a proteolytic enzyme that is involved in the degradation of periodontal connective tissue. Its increased level in saliva correlates with the inflammation of periodontium and peri-implant tissues, especially in the clinically active phase. The source of MMP-8 is neutrophils activated in response to plaque bacteria [17,18].

Apart from inflammation and degradation of connective tissue elements, another factor in the development of periodontal disease is the destruction of its bone structures, in which osteoclasts play a key role. 

The major regulator of osteoclast differentiation and function is RANKL (receptor activator of nuclear factor NF-kB ligand; RANKL). RANKL expression in periodontal tissues stimulates osteoclast activation through interaction with the RANK receptor. RANKL mRNA levels in periodontal tissue and RANKL protein in the gingival fluid are associated with the severity of periodontitis [19,20].

Osteoprotegerin (OPG) inhibits the resorption of bones or teeth by strongly binding to RANKL and thus blocking the activity of osteoclasts [21]. 

The disease process in the oral cavity coexists with an increase in peripheral blood inflammatory parameters, such as C-reactive protein (CRP) or the level of leukocytes [22,23,24]. The increase in the inflammatory parameters mentioned above is a predisposing factor to the occurrence of acute cerebral ischemic episodes [25].

Single scientific reports indicate the existence of a relationship between severe periodontitis and the severity of stroke [26]. To this day, it is not clear whether the occurrence of periodontal disease in patients with ischemic stroke affects the course of stroke. There are also no reports as to whether possible interventions to improve the condition of the oral cavity may play a role in the therapeutic process in these patients.

It is known that saliva plays a key role in maintaining a healthy oral cavity. This is due to the pleiotropic influence of saliva on moisturizing the oral cavity, decomposing sugars contained in food, antibacterial activity, cleaning the oral cavity by washing away sugars and acids from plaques and neutralizing acid production [27,28]. It is possible to use techniques to regulate salivation by massage of the submandibular and sublingual glands [29]. The stimulation of salivation helps to maintain its favorable physicochemical parameters [30,31,32].

The null hypotheses of the presented study state that the severity of periodontitis and the stimulation of saliva affect the course of acute phase of ischemic stroke in humans. The primary and secondary objectives of this investigation were to analyze the alternative hypothesis that periodontitis and the stimulation of saliva do not affect the course of acute phase of ischemic stroke in humans.

Therefore, the main aim of the study relied on the comprehensive assessment of the impact of periodontitis and the stimulation of salivation on the course of the acute phase of ischemic stroke in humans. The specific objectives of the study concerned the assessment of salivary inflammatory parameters depending on the severity of oral cavity inflammation as well as the search for a correlation between periodontitis and the severity and course of ischemic stroke, including the incidence of infectious complications in the acute phase of stroke. Finally, an attempt was made to assess the impact of neurologopedic therapy with massage of the oral salivary glands on the course of ischemic stroke in the context of the dynamics of symptoms and the occurrence of infectious complications.

## 2. Material and Methods

### 2.1. Materials

The study was designed as a prospective, open-label, nonrandomized clinical trial in a single center for subjects with a first ischemic stroke, with symptoms from the anterior cerebral artery (basin of the internal carotid artery), with a significant neurological deficit (minimum 3 points according to National Institute of Health Stroke Scale (NIHSS)). The study was approved (approval code: KB-0012/06/10; 25 January 2010) by the Ethics Committee of the Pomeranian Medical University in Szczecin (Poland) and performed in accordance with the Declaration of Helsinki. All patients provided written informed consent. 

The exclusion criteria were the following:-aphasia, disturbances of consciousness, mental disorders—making it impossible to express informed consent,-surgery of the salivary glands—disrupting the secretion of saliva,-diseases that disrupt salivary secretion (diabetes mellitus, Sjögren’s syndrome, condition after radiotherapy in the area of the salivary glands).

### 2.2. Methods

The diagnosis of ischemic stroke was made based on the medical history and a physical examination with assessment of the neurological status and a computed tomography (CT) scan of the head. The severity of stroke-related neurological symptoms was assessed using the NIHSS scale. The NIHSS results on admission (i.e., on the first day), on the third and seventh day of a stroke were analyzed.

As part of the assessment of the course of ischemic stroke during the patient’s stay in the ward, the dynamics of neurological symptoms were taken into account based on the NIHSS scale in subsequent days of hospitalization and the occurrence of infectious complications (upper respiratory tract infections, urinary tract infections, generalized bacterial infections). The diagnosis of the infection was based on the clinical picture (history, physical examination, temperature measurement), and the results of additional tests performed in everyday clinical practice in patients with stroke according to the indications (morphology, CRP, urinalysis, chest X-ray, procalcitonin, urine culture, culture of airway secretion, blood culture).

#### 2.2.1. Periodontal Condition Assessment

The condition of the oral cavity, including the periodontium, was assessed in each patient enrolled in the study. The examination was performed by the same, qualified dentist on the first day of the stroke, using a dental mirror and a calibrated periodontal probe, in daylight.

The following elements were assessed: missing teeth, the presence of active caries foci and the presence of existing fillings. As part of the periodontal examination, the following were assessed: the presence of dental deposits, the depth of the existing periodontal pockets, tooth mobility according to Hall (0—no mobility, 1—labio-lingual mobility up to 1 mm, 2—labio-lingual mobility up to 2 mm, 3—labio-lingual and vertical mobility), and bleeding during probing.

#### 2.2.2. Assessment of IL-1β, MMP-8, RANKL and OPG Expression in the Saliva of Patients with Ischemic Stroke

About 2 mL of saliva was collected twice (on the first and seventh day of stroke) from each of the patients qualified for the study in order to determine the concentration of IL-1β, MMP-8, RANKL, and OPG.

Initially, the saliva collected by spitting into test tubes was frozen at −70 °C and stored until the tests were performed.

Determination of IL-1β, RANKL and OPG expression was performed by the ELISA method using antibodies according to the protocol provided by the manufacturer, respectively: ELISA Kit for Interleukin-1β (ELISA Kit for Interleukin-1β, Wuhan EIAab Science, China), ELISA Kit for Human Tumor Necrosis Factor Ligand Superfamily Member 11 (TNFSF11) (ELISA Kit for Human Tumor Necrosis Factor Ligand Superfamily Member 11 (TNFSF11), Wuhan EIAab Science, Wuhan, China), Human TNFRSF11B/Tumor Necrosis Factor Receptor Superfamily Member 11B ELISA Kit (The Human Neutrophil Collagenase ELISA Kit, Wuhan EIAab Science, Wuhan, China).

MMP-8 concentration was determined by immunofluorescence according to the protocol provided by the manufacturer of the Human Neutrophil Collagenase ELISA Kit (The 2104 EnVision Multilabel Plate Reader, Perkin Elmer, Whaltham, MA, USA).

#### 2.2.3. Assessment of the Leukocyte Count and CRP Level in the Peripheral Blood of Patients with Ischemic Stroke

On days 1, 3 and 7 of ischemic stroke, the number of white blood cells (WBC) and the level of CRP in the peripheral blood of the patients were examined.

#### 2.2.4. Speech Therapy Massage

In each patient from the group subjected to stimulation of salivation, from the second day of hospitalization, every day until the end of hospitalization, a 15-min manual stimulation of the submandibular and sublingual glands was performed internally and externally. Each patient was stimulated seven times. Simultaneously with the stimulation, oral hygiene was carried out, taking into account the tongue and cheeks.

#### 2.2.5. Statistical Analysis

For the evaluation of quantitative variables, an arithmetic mean was used, taking into account the minimum and maximum values and the standard deviation. With reference to the analysis of qualitative variables, the percentage values and the number of the variable were used. Due to the distribution of the analyzed data deviating from the normal distribution (Shapiro–Wilk test, *p* <0.05), groups of variables were compared using non-parametric tests, employing the Mann–Whitney U-test for two groups. The Wilcoxon pairs order test was used to compare the dependent variables, and the Pearson’s Chi-squared test for 2 × 2 tables was used to analyze the qualitative variables. The significance level for the analyzed variables was *p* < 0.05. The licensed STATISTICA version 13.3 (StatSoft Inc., Kraków, Poland) program was used for statistical analysis.

## 3. Results

A total of 100 patients, 40 females and 60 males, between 46 and 75 years of age (66.1 ± 9.22) hospitalized at the Department of Neurology PUM in Szczecin on the first day of ischemic stroke were enrolled in the study. Patients who met the inclusion criteria were assigned alternately to the group undergoing neurological therapy (stimulation of the salivary glands and oral hygiene) (group I, *n* = 54) or to the group with no therapy (group II, *n* = 46). At the same time, the condition of the periodontium was examined in each patient. The division into groups is presented in Figure 1.

The characteristics of the patients enrolled in the study are presented in Table 1.

A detailed analysis of patients with a differentiation of the presence or absence of periodontal diseases is presented in Table 2.

Periodontal diseases occurred in 36% of the studied population. Initially (on the first day of stroke), the neurological status of patients with periodontal diseases was more severe than in other patients, although the differences were not statistically significant. On days three and seven of stroke, the condition of the patients without periodontal disease improved significantly when compared to patients with periodontal disease, *p* = 0.006 and *p* = 0.014, respectively. On the other hand, no statistically significant differences were found in the incidence of infection and the level of factors tested in the peripheral blood and in saliva between the groups.

Table 3 presents the differences in the level of selected inflammatory parameters in the peripheral blood and the concentration of IL-1β, RANKL, OPG and MMP-8 in saliva in the analyzed patients with and without periodontitis in the following days of stroke compared to the baseline values.

Next, patients with ischemic stroke were randomly assigned to neurologopedically stimulated and non-stimulated groups (Table 4).

Patients with ischemic stroke were randomly assigned to stimulated and non-stimulated neurologic groups. The groups did not differ statistically in age or gender distribution. Patients from the stimulated group had a more severe neurological deficit at baseline (*p* = 0.035). In the following days of neurological observation (+3 and +7), the condition of patients from both groups improved with a further distinct advantage of the unstimulated group over the stimulated group, *p* = 0.026 and *p* = 0.0004, respectively. There were no significant differences in the incidence of infection and in the level of factors tested in peripheral blood and in saliva between the groups on different days of the stroke.

Table 5 presents the differences in the level of CRP and the number of peripheral blood leukocytes as well as the concentrations of IL-1β, RANKL, OPG and MMP-8 in the saliva of patients stimulated and unstimulated by speech therapy in the following days of stroke when compared to the baseline values.

In patients from both groups, a statistically significant decrease in CRP and the number of leukocytes was observed on day seven in relation to day one.

The concentrations of the factors tested in saliva in both groups on the seventh and the first day were comparable.

## 4. Discussion

Periodontal diseases occur in 20–85% of the population [33,34]. In the analyzed group, they were found in 36% of patients. In our study group, among patients with periodontitis, there were patients with periodontitis as well as edentulous people. Toothless persons (20 patients) were taken into account, bearing in mind that tooth loss is a marker of a long-term periodontitis, indicates the health of the oral cavity and is pathogenetically associated with the risk of ischemic stroke [26,35,36].

The research suggests that the presence of these conditions increases the risk of ischemic stroke [36,37,38,39]. Whether their occurrence affects the course of a stroke remains an open question.

In the presented material, patients with periodontal disease were initially characterized by a greater neurological deficit. The neurological status of all patients improved in the following days of stroke, but the differences between the groups were statistically significant in favor of patients without periodontal disease. Is it possible then to conclude that the occurrence of periodontal disease contributes to a worse course of stroke?

Single reports refer to the influence of periodontal disease on the severity of stroke and post-stroke disability [26,39,40]. Słowik et al. showed that patients with periodontal diseases (including toothless) had a more severe neurological status at baseline when compared to patients without periodontal diseases [26]. Leira and associates published results suggesting that periodontitis adversely affects post-stroke disability [39]. There are no studies in the available literature relating periodontal disease to the dynamics of symptoms associated with stroke.

When discussing the course of the acute phase of stroke, attention should be paid to the presence of infection. Poor oral health in stroke patients may be one of the factors that increases their risk of respiratory infections [41,42,43].

In the analyzed patients, the occurrence of infection was observed more often in patients with periodontal disease. These were infections of the respiratory tract, although the differences were not statistically significant. Boaden et al. also observed that poor oral health in stroke patients is associated with infections, predominantly respiratory ones [41]. A similar trend was observed by Cieplik et al., in a prospective study assessing the health and bacterial flora of the oral cavity of patients admitted to the stroke department. The researchers found that the incidence of pneumonia in patients with confirmed stroke was more frequent in patients with larger dental cavities and poorer oral hygiene [44].

An analysis of a larger group of patients could change the strength of our observation. However, in the light of the available literature data, oral care in patients with acute stroke appears to be one of the elements of the prevention of respiratory tract infections in these patients.

It is known that a generalized inflammatory reaction may intensify local inflammatory processes in the area of acute focal ischemia in the brain [45,46].

The assessment of CRP concentration and the number of leukocytes in the blood of the analyzed patients on the first, third and seventh day of stroke may indicate a greater intensity of inflammatory processes in patients with periodontal disease than in other patients. However, these differences were not statistically significant. These groups did not differ significantly in terms of biochemical parameters reflecting the severity of periodontal disease, except for a slight advantage in the concentration of pro-inflammatory IL-1β in patients with periodontal disease. This may be due to the fact that patients with periodontitis did not have active periodontitis. The available studies suggest that IL-1β levels are higher at the time of active inflammation when compared to the time of remission [47,48,49].

O’Boyle et al. demonstrated in an animal model that periodontitis results in an increase of inflammatory parameters in the blood. However, they did not find that this mattered to the size of the stroke or damage to the blood–brain barrier. They also did not observe that the occurrence of periodontitis in mice was significant for the inflammatory parameters in the blood after the induction of ischemic stroke [40].

In the present study, in patients with periodontal disease, as in patients without periodontal disease, a significant decrease in CRP and the number of leukocytes in the blood were observed on the seventh day of stroke when compared to the first one. The neurological status of patients with periodontal disease, although worse than in patients without periodontal disease, improved in the following days of stroke. Perhaps the role of periodontal disease in stroke should not be overestimated, or maybe the intervention in the form of stimulation of the salivary glands and the oral cavity, in addition to the standard therapy, could have had an impact on the observed favorable dynamics of neurological symptoms and the results of inflammatory parameters in the blood in the analyzed patients.

The stimulation of saliva secretion is conducive to the maintenance of saliva parameters favorable for the oral health [50]. The use of neurologopedic therapy in a patient in the acute phase of stroke may reduce the risk of developing upper and lower respiratory tract infections by maintaining proper moisturization of the oral cavity and the antibacterial activity of saliva [51].

In the sub-acute phase of stroke, a dysfunction of the salivary gland occurs. During stimulation, the parotid gland produces up to 60% of saliva [52]. Therefore, post-stroke patients suffer from dysfunction mainly of the parotid glands [53].

In the presented study, 54 patients with stroke underwent manual stimulation. In the group of patients with periodontal disease, their proportion was slightly over 61%. In the group of patients without periodontal disease, they constituted half.

The stimulated and unstimulated patients were comparable in terms of age and sex. However, the stimulated ones differed from the others with a significantly greater intensity of the neurological deficit. Already at the beginning, their neurological condition was worse. The trend of more frequent infections, more frequent use of antibiotics, and higher blood CRP when compared to unstimulated patients should be associated with de facto more severe stroke rather than the use of stimulation [26].

However, in the following days of stroke, an improvement in the neurological status of stimulated patients was observed and a favorable decrease in inflammatory parameters on the seventh day of stroke compared to the baseline was noted. It cannot be ruled out that the use of stimulation in these patients translated into such dynamics of the discussed parameters.

In the available studies, different parameters of stimulated and unstimulated saliva were measured. There was no significant translation of the stimulation of salivary secretion into the biochemical parameters of saliva, indicating the intensification of the mechanisms involved in the pathogenesis of periodontitis. A study using Illumina sequencing showed that samples of stimulated and unstimulated saliva (drooling) from the same subjects were not statistically significant [53,54,55]. Belstrøm et al. found that the stimulated saliva samples showed completely different bacterial profiles when compared to the subgingival samples [54]. Unstimulated and stimulated total saliva had similar concentrations of testosterone, androstenedione, and cortisol [56]. In each of these studies, no evident improvement in selected parameters in the saliva was demonstrated before and after stimulation. Thus, one should consider what is the reason for this. Most likely, this may be due to the method itself, as well as the time of saliva collection for testing [57,58].

### 4.1. Study Limitations

The weak point of the study is the random selection of groups subjected or not subjected to the application of the discussed stimulation of the oral cavity and salivary glands. Differences in the baseline stroke severity made it difficult to interpret the results obtained. It is worth noting, however, that the frequency of infectious complications in stimulated patients, despite their more severe neurological condition, was not significantly higher than in other patients. The same was true for the inflammatory parameters in the blood.

### 4.2. Conclusions for the Future

(1)The test and control groups should be properly selected, at least in terms of age and sex, and possibly also in terms of comorbidities.(2)There is a need for the research on a larger group of patients.(3)The panel of biochemical tests should be expanded to include CRP in saliva, and the concentration of IL-1β, OPG, RANKL and MMP-8 in the blood, which would allow for a discussion of the results in saliva in the context of the analyzed role of periodontal disease in the course of stroke.(4)The collection method should be developed and the collection time most favorable for the evaluation of saliva parameters should be determined.

## 5. Conclusions

The null hypotheses that periodontal disease and saliva stimulation may influence the course of ischemic stroke has not been unequivocally confirmed. However, it has been shown that:The occurrence of periodontal disease in a patient with stroke affects the severity of stroke.Ttimulation of the mouth and salivary glands in these patients may have a positive effect on the course of stroke, taking into account the dynamics of neurological symptoms.

## Figures and Tables

**Figure 1 jcm-11-04321-f001:**
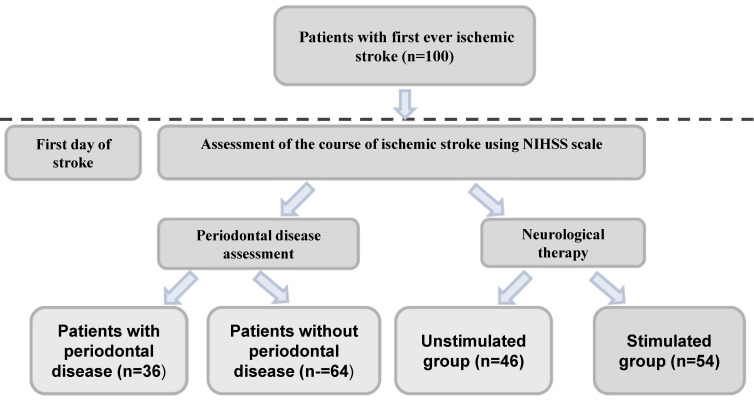
Division of patients with ischemic stroke into specific groups. NIHS—National Institute of Health Stroke Scale.

**Table 1 jcm-11-04321-t001:** General characteristics of the examined population.

Parameters		All Patients (*n* = 100)
Demographic data		
Age (years; mean ± SD)		66.1 ± 9.22
Sex	Females	40 (40%)
	Males	60 (60%)
Periodontal disease		36 (36%)
**Characteristics of the groups**		
Neurologopedic stimulation	Stimulated patients	54 (54%)
	Unstimulated patients	46 (46%)
**Neurological state**		
NIHSS scale (points; mean ± SD)	1st day	7.34 ± 4.76
	3rd day	4.8 ± 4.79
	7th day	3.64 ± 5.14
**Infections**		
Antibiotic therapy		38 (38%)
Upper respiratory tract infection		23 (23%)
Urinary tract infection		14 (14%)
**Inflammatory parameters from blood**		
CRP (mg/L; mean ± SD)	1st day	9.12 ± 1.52
	3rd day	1.49 ± 3.63
	7th day	0.7 ± 1.54
WBC (G/µL; mean ± SD)	1st day	8.54 ± 3.20
	3rd day	8.48 ± 3.44
	7th day	7.75 ± 2.99
**Concentration of biomarkers of inflammation, bone remodeling and degradation of connective tissue elements in saliva**		
IL-1β (pg/mL; mean ± SD)	1st day	28.9 ± 31.08
	7th day	22.45 ± 19.80
RANKL (ng/mL; mean ± SD)	1st day	0.28 ± 0.06
	7th day	0.27 ± 0.05
OPG (ng/mL; mean ± SD)	1st day	0.45 ± 0.33
	7th day	0.4 ± 0.24
MMP-8 (ng/mL; mean ± SD)	1st day	3524 ± 2466.10
	7th day	3744.4 ± 2583.69

NIHSS, National Institute of Health Stroke Scale; SD, standard deviation; MMP-8, metalloproteinase-8; CRP, C-reactive protein; RANKL, receptor activator of nuclear factor NF-kB ligand; OPG, Osteoprotegerin; IL-1β, Interleukin 1β. WBC, Leukocytes.

**Table 2 jcm-11-04321-t002:** Characteristics of stroke patients treated with and without periodontal disease.

Parameters		Patients with Periodontal Diseases (*n* = 36)	Patients without Periodontal Diseases (*n* = 64)	*p*-Value
Demographic data				
Age (years; mean ± SD)		64.86 ± 8.99	66.79 ± 9.34	0.43
Sex	females	15 (41.67%)	25 (39.06%)	0.79
	males	21(58.33%)	39 (60.94%)	
**Neurological state**				
NIHSS scale (points; mean ± SD)	1st day	8.88 ± 5.97	6.46 ± 3.71	0.08
	3rd day	7.00 ± 6.11	3.56 ± 3.32	0.006 *
	7th day	5.94 ± 6.99	2.34 ± 3.12	0.014 *
**Infections**				
Antibiotic therapy	no	20 (55.56%)	42 (65.63%)	0.32
	yes	16 (44.44%)	22 (34.38%)	
Upper respiratory tract infection	no	27 (75%)	50 (78.13%)	0.72
	yes	9 (25%)	14 (21.88%)	
Urinary tract	no	29 (80.5%)	57 (89.06%)	0.23
Infection	yes	7 (19.44%)	7 (10.94%)	
**Assessment of inflammatory parameters from blood**				
CRP (mg/L; mean ± SD)	1st day	10.72 ± 19.07	8.21 ± 12.67	0.26
	3rd day	15.83 ± 27.11	14.47 ± 40.82	0.15
	7th day	9.44 ± 15.95	6.6 5± 15.14	0.8
WBC (G/µL; mean ± SD)	1st day	8.52 ± 1.73	8.56 ± 3.81	0.31
	3rd day	8.87 ± 2.99	8.26 ± 3.67	0.24
	7th day	7.99 ± 3.14	7.62 ± 2.92	0.56
**Concentration of biomarkers of inflammation, bone remodeling and degradation of connective tissue elements in saliva**				
IL-1β (pg/mL; mean ± SD)	1st day	25.46 ± 19.71	30.83 ± 35.93	0.58
	7th day	26.29 ± 26.81	20.70 ± 14.53	0.57
RANKL (ng/mL; mean ± SD)	1st day	0.29 ± 0.07	0.28 ± 0.06	0.43
	7th day	0.28 ± 0.08	0.28 ± 0.06	0.15
OPG (ng/mL; mean ± SD)	1st day	0.49 ± 0.42	0.42 ± 0.27	0.96
	7th day	0.44 ± 0.25	0.39 ± 0.23	0.24
MMP-8 (ng/mL; mean ± SD)	1st day	3371.63 ± 2384.37	3604.18 ± 2525.60	0.66
	7th day	4155.31 ± 2626.78	3577.85 ± 2557.43	0.3
**Neurologopedic stimulation**				
	stimulated patients	22 (61.11%)	32 (50%)	0.28

NIHSS, National Institute of Health Stroke Scale; SD, standard deviation; MMP-8, metalloproteinase-8; CRP, C-reactive protein; RANKL, receptor activator of nuclear factor NF-kB ligand; OPG, Osteoprotegerin; IL-1β, Interleukin 1β; WBC, Leukocytes; * *p* < 0.05.

**Table 3 jcm-11-04321-t003:** The differences between the following time points and day 1 (baseline) of CRP level and white blood cells in the blood and RANKL, OPG, IL-1β and MMP-8 concentration in saliva in the group of patients with and without periodontal diseases.

Variables	Patients with Periodontal Diseases (*n* = 36)	Patients without Periodontal Diseases (*n* = 64)
	*p*-Value	*p*-Value
Assessment of inflammatory parameters from blood		
CRP day 3 vs. CRP day 1	0.682	0.057
CRP day 7 vs. CRP day 1	**0.009 ***	**0.0002 ***
WBC day 3 vs. WBC day 1	0.887	0.169
WBC day 7 vs. WBC day 1	**0.010 ***	**0.019 ***
**Concentration of biomarkers of inflammation, bone remodeling and degradation of connective tissue elements in saliva**		
IL-1β day 7 vs. IL-1Β day 1	0.65	**0.04 ***
RANKL day 7 vs. RANKL day 1	0.27	0.89
OPG day 7 vs. OPG day 1	1.00	0.68
MMP-8 day 7 vs. MMP-8 day 1	0.16	0.88

NIHSS, National Institute of Health Stroke Scale; SD, standard deviation; MMP-8, metalloproteinase-8; CRP, C-reactive protein; RANKL, receptor activator of nuclear factor NF-kB ligand; OPG, Osteoprotegerin; IL-1β, Interleukin 1β; WBC, Leukocytes; * *p* < 0.05.

**Table 4 jcm-11-04321-t004:** Characteristics of stroke patients treated with and without neurological stimulation.

Parameters		Patients with Neurologopedic Stimulation (*n* = 54)	Patients without Neurologopedic Stimulation (*n* = 46)	*p*-Value
Demographic data				
Age (years; mean ± SD)		66.00 ± 9.90	66.21 ± 8.45	0.73
Sex	Females	21 (38.8%)	19 (41.3%)	0.8
	Males	33 (61.1%)	27 (58.6%)	
**Neurological state**				
NIHSS scale (points; mean ± SD)	1st day	8.25 ± 4.82	6.26 ± 4.51	**0.035 ***
	3rd day	5.64 ± 4.91	3.80 ± 4.50	**0.026 ***
	7th day	5.04 ± 5.52	2.00 ± 4.15	**0.0004 ***
**Infections**				
Antibiotic therapy	No	30 (55.5%)	32 (69.5%)	0.15
	Yes	24 (44.5%)	14 (30.5%)	
Upper respiratory tract infection	No	40 (74.07%)	37 (80.4%)	0.45
	Yes	14 (25.9%)	9 (19.5%)	
Urinary tract infection	No	44 (81.4%)	42 (91.3%)	0.15
	Yes	10 (18.5%)	4 (8.6%)	
**Inflammatory parameters from blood**				
CRP (mg/L; mean ± SD)	1st day	10.24 ± 16.97	7.79 ± 12.98	0.35
	3rd day	20.15 ± 45.66	8.87 ± 19.51	0.21
	7th day	9.78 ± 17.63	5.16 ± 12.04	0.23
WBC (G/µL; mean ± SD)	1st day	8.99 ± 3.83	8.24 ± 2.20	0.2
	3rd day	9.01 ± 4.26	7.86 ± 1.99	0.28
	7th day	8.04 ± 3.60	7.41 ± 2.04	0.79
**Concentration of biomarkers of inflammation, bone remodeling and degradation of connective tissue elements in saliva**				
IL-1β (pg/mL; mean ± SD)	1st day	30.71 ± 37.81	26.77 ± 20.97	0.94
	7th day	24.55 ± 22.28	20.51 ± 16.47	0.14
RANKL (ng/mL; mean ± SD)	1st day	0.29 ± 0.07	0.28 ± 0.04	0.45
	7th day	0.28 ± 0.06	0.27 ± 0.02	0.4
OPG (ng/mL; mean ± SD)	1st day	0.47 ± 0.39	0.42 ± 0.26	0.99
	7th day	0.44 ± 0.28	0.35 ± 0.16	0.23
MMP-8 (ng/mL; mean ± SD)	1st day	3498.34 ± 2519.39	3546.44 ± 2429.53	0.92
	7th day	3708.37 ± 2656.80	3866.84 ± 2523.34	0.8

NIHSS, National Institute of Health Stroke Scale; SD, standard deviation; MMP-8, metalloproteinase-8; CRP, C-reactive protein; RANKL, receptor activator of nuclear factor NF-kB ligand; OPG, Osteoprotegerin; IL-1β, Interleukin 1β; WBC, Leukocytes; * *p* < 0.05.

**Table 5 jcm-11-04321-t005:** The differences between the following time points and day 1 (baseline) of CRP level and white blood cells in the blood and Il-1β, RANKL, OPG and MMP-8 concentration in saliva in the group of patients with and without neurologopedic stimulation.

Variables	Patients with Neurologopedic Stimulation (*n* = 54)	Patients without Neurologopedic Stimulation (*n* = 46)
	*p*-Value	*p*-Value
Assessment of inflammatory parameters from blood		
CRP day 3 vs. CRP day 1	0.29	0.15
CRP day 7 vs. CRP day 1	**0.0065 ***	**0.0001 ***
WBC day 3 vs. WBC day 1	0.53	0.47
WBC day 7 vs. WBC day 1	**0.007 ***	**0.047 ***
**Concentration of biomarkers of inflammation, bone remodeling and degradation of connective tissue elements in saliva**		
IL-1Β day 7 vs. IL-1Β day 1	0.69	0.10
RANKL day 7 vs. RANKL day 1	0.22	0.94
OPG day 7 vs. OPG day 1	0.45	0.21
MMP-8 day 7 vs. MMP-8 day 1	0.86	0.37

NIHSS, National Institute of Health Stroke Scale; SD, standard deviation; MMP-8, metalloproteinase-8; CRP, C-reactive protein; RANKL, receptor activator of nuclear factor NF-kB ligand; OPG, Osteoprotegerin; IL-1β, Interleukin 1β; WBC, Leukocytes; * *p* < 0.05.

## Data Availability

The data presented in this study are available in [Department of Neurology, Pomeranian Medical University, Szczecin, Poland].

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
