# Peer review of "The Influence of Periodontal Diseases and the Stimulation of Saliva Secretion on the Course of the Acute Phase of Ischemic Stroke"

_jcm, 2022, doi:10.3390/jcm11154321_

Round 1
Reviewer 1 Report
Dear Authors,
The specific objectives of the study concerned the assessment of salivary inflammatory parameters depending on the severity of oral cavity inflammation as well as the search for a correlation between periodontitis and the severity and course of ischemic stroke, including the incidence of infectious complications in the acute phase of stroke.
The study is of scientific interest and in line with the aims of the journal. The author guidelines have been respected.
However, there are some issues that should be addressed.
Abstract
Abstract: The abstract should be a total of about 200 words maximum. The abstract should be a single paragraph and should follow the style of structured abstracts, but without headings: 1) Background: Place the question addressed in a broad context and highlight the purpose of the study; 2) Methods: Describe briefly the main methods or treatments applied. Include any relevant preregistration numbers, and species and strains of any animals used. 3) Results: Summarize the article's main findings; and 4) Conclusion: Indicate the main conclusions or interpretations. The abstract should be an objective representation of the article: it must not contain results which are not presented and substantiated in the main text and should not exaggerate the main conclusions. (https://www.mdpi.com/journal/jcm/instructions)
- Lines 16-17. “100 consecutive patients with their first ever ischemic stroke were enrolled in the study. 56 randomly selected patients were subjected to stimulation of salivation”. Put numbers information in the Results and not in the material and methods.
- Among keywords put “;” e not “,”.
Introduction
- Please refresh the link between periodontal disease and some frequent systemic pathologies. Please cite doi: 10.3390/jcm10194578.
- Please clarify the difference between gingivitis and periodontitis. I suggest to clarify that plaque biofilm is considered as etiological factor in the onset of gingivitis and it is implicated in the progression to periodontitis and peri-implant inflammation. It is important to highlight that host and environmental factors significantly contribute to the advancement of the inflammatory process.
- Line 52-53. “It has been proven that poor oral health may be an additional factor that increases 52 susceptibility to infections in patients with stroke”. Please cite doi: 10.1080/10749357.2019.1673592.
Material and methods
- Lines 125-127. “A total of 100 patients, 40 females and 60 males, between 46 and 75 years of age (66.1 ± 9.22) hospitalized at the Department of Neurology PUM in Szczecin on the first day of ischemic stroke were enrolled in the study”. You have to put this information in the Result Section.
- - Lines 134-135. “Patients who met the inclusion criteria were assigned alternately to the group under going neurological therapy (stimulation of the salivary glands and oral hygiene) (group I, n = 54) or to the group with no therapy (group II, n = 46).” You have to put this information in the Result Section.
- Line 186. “(n = 54)”. Put this information in the Result Section.
The Result and Discussion sections were well described.
References
The references have to be written as follow:
Journal Articles:
1. Author 1, A.B.; Author 2, C.D. Title of the article. Abbreviated Journal Name Year, Volume, page range.
. (https://www.mdpi.com/journal/jcm/instructions)
Author Response
Szczecin, 13 July 2022
Journal of Clinical Medicine
Editors
Manuscript number: 1802067
“The influence of periodontal diseases and the stimulation of saliva secretion on the course of the acute phase of ischemic stroke” by Wioletta Pawlukowska, BartĹ‚omiej Baumert, Agnieszka Meller, Anna Dziewulska, Alicja ZawiĹ›lak, Katarzyna Grocholewicz, PrzemysĹ‚aw Nowacki, Masztalewicz Marta.
We were pleased to read the constructive comments of the Reviewers and their suggestion that the manuscript could be considered for publication in the Journal, with some major revision. We reworked and corrected our paper according to the Reviewers’ requests, performing changes in main manuscript. In revised manuscript all changes are indicated using the editing tools. In response to Reviewers, all the changes are indicated in red. We trust that the revised version of our manuscript is clearer and strengthened scientifically. We thank the Reviewers for their comments and careful evaluation of our paper. We are happy to address all the Reviewers comments point by point below.
We hope it will now meet with your approval for publication in Journal of Clinical Medicine. Thank you for your time and I am looking forward to hearing from you.
Sincerely yours,
Corresponding author:
Wioletta Pawlukowska, PhD, DSc
Department of Neurology,
Pomeranian Medical University in Szczecin
Unii Lubelskiej 1
71-252 Szczecin, Poland
phone: +48 914800914
fax: +48 914800918
e-mail: wioletta.pawlukowska@pum.edu.pl
-------------------------------------------Reviewer #1’s Comments -------------------------------------
Dear Reviewer,
Thank you for your comments and kind opinion concerning our manuscript entitled
”The influence of periodontal diseases and the stimulation of saliva secretion on the course of the acute phase of ischemic stroke”. We have studied the comments carefully and have made corrections, which we hope, will meet with your approval.
Points of criticism:
-
Lines 16-17. “100 consecutive patients with their first ever ischemic stroke were enrolled in the study. 56 randomly selected patients were subjected to stimulation of salivation”. Put numbers information in the Results and not in the material and methods. Among keywords put “;” e not “,”.
(The response)
We moved the sentence “100 consecutive patients with their first ever ischemic stroke were enrolled in the study. 56 randomly selected patients were subjected to stimulation of salivation” to the results section. In the keywords we changed "," to ";".
Points of criticism:
2) Introduction
- Please refresh the link between periodontal disease and some frequent systemic pathologies. Please cite doi: 10.3390/jcm10194578.
- Please clarify the difference between gingivitis and periodontitis. I suggest to clarify that plaque biofilm is considered as etiological factor in the onset of gingivitis and it is implicated in the progression to periodontitis and peri-implant inflammation. It is important to highlight that host and environmental factors significantly contribute to the advancement of the inflammatory process.
-Line 52-53. “It has been proven that poor oral health may be an additional factor that increases 52 susceptibility to infections in patients with stroke”. Please cite doi: 10.1080/10749357.2019.1673592.
(The response)
We supplemented the missing information and citations as required.
Points of criticism:
3) Material and methods
- Lines 125-127. “A total of 100 patients, 40 females and 60 males, between 46 and 75 years of age (66.1± 9.22) hospitalized at the Department of Neurology PUM in Szczecin on the first day of ischemic stroke were enrolled in the study”. You have to put this information in the Result Section.
-Lines 134-135. “Patients who met the inclusion criteria were assigned alternately to the group undergoing neurological therapy (stimulation of the salivary glands and oral hygiene) (group I, n = 54) or to the group with no therapy (group II, n = 46).” You have to put this information in the Result Section.
-Line 186. “(n = 54)”. Put this information in the Result Section.
(The response)
We moved the indicated elements to the results section as required.
Points of criticism:
4) References
The references have to be written as follow:
Journal Articles:
1. Author 1, A.B.; Author 2, C.D. Title of the article. Abbreviated Journal Name Year, Volume, page range.
(The response)
The references have been corrected as indicated.
We would like to thank the Reviewers for their helpful comments and hope that our manuscript is now more balanced, clearer, and better represents our work. We hope that the revised manuscript is now acceptable for publication in Journal of Clinical Medicine.
Reviewer 2 Report
Dear Authors,
Thank you for submitting you valuable work to the journal. The pathogenic connections between Periodontitis and Cardiovascular Diseases have received a lot of scientific attention, given their significant impact of the patient's life. However, the study of the periodontal status in patients with storke is less common, given the clinical difficulties these patients encounter. Thus, your work is of great interest and the topic of the study is highly valuable. Nevertheless, I would make some comments in order to increase the paper's scientific accuracy:
- The Introduction is too wordy and long, it should be reorganized and essential information extracted
- A Null hypothesis should be added to the Objectives
- Study Group composition and charcterristics should be more clearly exhibited
- Discussions are too long and wordy, a reorganization of the included information should be performed for improved understanding of the delivered message and significance of the results
- Points 5.1 and 5.2 should be moved to the End of the Discussion Section
- Please rephrase Conclusion in accordance to Objectives and Null Hypothesis
We look forward to receiving the revised version of your manuscript!
Kind regards!
Author Response
Dear Reviewer,
Thank you for your comments and kind opinion concerning our manuscript entitled “The influence of periodontal diseases and the stimulation of saliva secretion on the course of the acute phase of ischemic stroke”. We have studied the comments carefully and have made corrections, which we hope, will meet with your approval.
Points of criticism:
-
The Introduction is too wordy and long, it should be reorganized and essential information extracted.
(The response)
According to the Reviewer suggestion we have shortened the Introduction section. We left behind what we think is necessary to understand the topic.
Points of criticism:
-
A null hypothesis should be added to the Objectives.
(The response)
We added null hypothesis as required.
Points of criticism:
-
Study Group composition and characteristics should be more clearly exhibited.
(The response)
We have added Figure 1 for a better explanation of the group division.
Points of criticism:
-
Discussions are too long and wordy, a reorganization of the included information should be performed for improved understanding of the delivered message and significance of the results.
(The response)
According to the Reviewer suggestion we have shortened the Discussion section and slightly modified the Conclusion section.
Points of criticism:
-
Points 5.1 and 5.2 should be moved to the End of the Discussion Section.
(The response)
We moved the selected points to the indicated section.
Points of criticism:
-
Please rephrase Conclusion in accordance to Objectives and Null Hypothesis.
(The response)
We added information on the results of the null hypothesis in the conclusions.
We would like to thank the Reviewers for their helpful comments and hope that our manuscript is now more balanced, clearer, and better represents our work. We hope that the revised manuscript is now acceptable for publication in Journal of Clinical Medicine.
Round 2
Reviewer 1 Report
Authors have modified the text according to the suggestions, performing changes in main manuscript. In revised manuscript all changes are indicated using the editing tools.
Reviewer 2 Report
Dear authors,
Thank you for submitting the revised version of your manuscript. All comments have been answered accordingly, the paper increasing its scientific accuracy and impact.
I have no further suggestions to make.
Kind regards